# Environmental Enrichment and Metformin Improve Metabolic Functions, Hippocampal Neuron Survival, and Hippocampal-Dependent Memory in High-Fat/High-Sucrose Diet-Induced Type 2 Diabetic Rats

**DOI:** 10.3390/biology12030480

**Published:** 2023-03-21

**Authors:** Teh Rasyidah Ismail, Christina Gertrude Yap, Rakesh Naidu, Narendra Pamidi

**Affiliations:** 1Jeffrey Cheah School of Medicine and Health Sciences, Monash University Malaysia, Jalan Lagoon Selatan, Bandar Sunway, Subang Jaya 47500, Selangor Darul Ehsan, Malaysiachristina.yap@monash.edu (C.G.Y.);; 2Clinical Laboratory Science Section, Institute of Medical Science Technology, Universiti Kuala Lumpur, Kajang, Kuala Lumpur 43000, Selangor Darul Ehsan, Malaysia

**Keywords:** type 2 diabetes, memory impairment, environmental enrichment, metformin, hippocampus, BDNF-TrkB signaling pathways

## Abstract

**Simple Summary:**

Type 2 diabetes can lead to catastrophic complications, including neurodegeneration and memory impairment. Therefore, it is essential to identify effective therapeutic strategies to improve blood glucose levels and prevent the onset of complications. This study evidently showcases environmental enrichment as an effective therapy for preserving mental health in diet-induced type 2 diabetic rats. The outcomes of this study can be translated to clinical trials in diabetic patients. Environmental enrichment can be introduced as one of the alternative therapies for preventing diabetes in pre-diabetic individuals and in established diabetes alongside metformin or other hypoglycemic pharmacotherapy.

**Abstract:**

**Background**: The Western-style diet-induced type 2 diabetes mellitus (T2D) may eventually trigger neurodegeneration and memory impairment. Thus, it is essential to identify effective therapeutic strategies to overcome T2D complications. This study aimed to investigate the effects of environmental enrichment (EE) and metformin interventions on metabolic dysfunctions, hippocampal neuronal death, and hippocampal-dependent memory impairments in high-fat/high-sucrose (HFS) diet-induced T2D rats. **Methods:** Thirty-two male rats (200–250 g) were divided into four groups: C group (standard diet + conventional cage); D group (HFS diet + conventional cage); DE group (HFS diet + EE cage/6hr daily); and DM group (HFS diet + metformin + conventional cage). Body weight was measured every week. T-maze tasks, anthropometric, biochemical, histological, and morphometric parameters were measured. The expression changes of hippocampal genes were also analyzed. **Results:** The anthropometric and biochemical parameters were improved in DE and DM groups compared with the D group. DE and DM groups had significantly higher T-maze percentages than the D group. These groups also had better histological and morphometric parameters than the D group. The interventions of EE and metformin enhanced the expression of hippocampal genes related to neurogenesis and synaptic plasticity (BDNF/TrkB binding, PI3K-Akt, Ras–MAPK, PLCγ–Ca^2+^, and LTP). **Conclusion:** Environmental enrichment (EE) and metformin improved metabolic functions, hippocampal neuron survival, and hippocampal-dependent memory in HFS diet-induced T2D rats. The underlying mechanisms of these interventions involved the expression of genes that regulate neurogenesis and synaptic plasticity.

## 1. Introduction

Type 2 diabetes mellitus (T2D) contributes to the worldwide disease burden, and devastatingly, the global prevalence of T2D is projected to increase to 7079 individuals per 100,000 by 2030 [1]. T2D is often initiated by obesity, a disease caused by the overconsumption of a Western-style diet [2]. The most representative examples of Western-style diets are fast foods, high-fat foods, and sugar-sweetened desserts and beverages. Studies on the effects of western-style diets on animals were performed using modified or enriched diets consisting of high percentages of fat and sucrose [2].

Over time, T2D can trigger a critically long-lasting impact on the hippocampus and its role in memory formation. The pathophysiology linking T2D with hippocampal impairments involved the combination of several factors. These factors include but are not limited to hyperglycemia, insulin resistance, adipose tissue dysfunction, and oxidative stress [3]. For this reason, effective therapeutic strategies are essential to overcome T2D-related hippocampal impairments. Some studies even emphasized the medicine-free approach for the interventions of these impairments [4]. In animal models, investigations on the medicine-free approach can be conducted using environmental enrichment (EE).

Environmental enrichment (EE) refers to complex motor stimulation and sensory and cognitive enhancement that provides animals with physical exercise opportunities, entertaining activities, various learning experiences, and complex social interactions [5]. EE cage is a large-space cage equipped with rotating or running wheels, toys, tubes, and tunnels. The accessibility of rotating or running wheels results in more voluntary exercise. Simultaneously, repeated exposures to novel items, such as toys, tubes, and tunnels, increase the opportunity to experience new sensory information [6]. Previous studies showed that EE enhanced learning and memory [7]. Some reports demonstrated that EE increased hippocampal neurogenesis and enhanced the expression of neurotrophic factors [8].

Biguanide metformin is a safe, oral anti-hyperglycemic drug of choice to treat T2D in obese and non-obese patients. Furthermore, it is inexpensive and has a good safety profile. Metformin has antioxidant properties that are capable of reducing oxidative stress and neuroinflammation. Thus, it has also recently attracted much attention due to its possible beneficial effects on the learning and memory [9].

The EE effects on various disease models are well-documented, and the anti-diabetic effect of metformin is well-known. However, the impact of EE and metformin against hippocampal impairments in HFS diet-induced T2D models are limited. Therefore, the present study was designed to investigate the effects of EE and metformin interventions on metabolic dysfunctions, hippocampal neuronal death, and hippocampal-dependent memory impairments in high-fat/high-sucrose (HFS) diet-induced T2D rats.

## 2. Materials and Methods

### 2.1. Animals and Ethics Statement

Thirty-two healthy male Wistar albino rats (200–250 g body weight) were obtained from the Jeffrey Cheah School of Medicine and Health Sciences Animal Facility, Monash University, Malaysia. All animal experiments were approved by the Monash University Animal Ethics Committee (AEC No. MARP/2016/109). Rats were acclimatized (1 week) before the study started. Rats were kept in conventional polypropylene cages (length: 460 mm × width: 300 mm × height: 160 mm) and were placed in a temperature-maintained (22–25 °C) and light-controlled (12-h/12-h light/dark phase) room. Rats had ad libitum access to standard chow pellets and tap water.

### 2.2. Diets

Standard and HFS diets were used in this study. A standard chow pellet diet comprised 20% protein, 59.4% carbohydrate, 4.8% fat, and 4.8% crude fiber with a caloric density of approximately 3.34 kcal/g (Specialty feeds, Glen Forrest, WA, Australia). The T2D was induced in the rats by feeding them a high-fat diet and a high-sucrose solution drink (HFS). The high-fat diet was composed of 20% protein, 20% carbohydrate, and 60% fat, with a caloric density of approximately 5.24 kcal/g (D12492; Research Diets, New Brunswick, NJ, USA). The 10% *w*/*v* sucrose solution in tap water was made fresh daily to drink ad libitum. The HFS diet mimicked the Western-style diet consisting of high-fat food and sweetened drinks [2].

### 2.3. Experimental Design

An illustration of the timeline of the experimental design is shown in Figure 1. Following one week of acclimatization, baseline parameters, such as body weight, BMI, biochemical parameters, and spatial learning and memory T-maze tasks, were measured. After completing the T maze tasks, eight rats were randomly allocated to the control group (C group, *n* = 8) on week two. These rats were placed in conventional polypropylene cages and were maintained on an ad libitum standard chow pellet diet and tap water. In contrast, the remaining twenty-four rats were fed the HFS diet to induce T2D. The HFS diet feeding procedure lasted ten weeks, followed by fasting blood glucose (FBG) measurement. Rats with fasting blood glucose above 6.7 mmol/L suggested that T2D was successfully induced and were kept in the study [10]. These rats were randomly subdivided into three groups of eight rats (D, DE, and DM groups) and were continuously fed the HFS diet until the end of the study. As interventions, rats in the DE group were placed in EE cages for 6 h/day, and rats in the DM group were given the anti-diabetic drug metformin by gavage feeding at a dose of 250 mg/kg body weight once daily. Bodyweight measurement was taken every week, and the T-maze tasks were conducted again in week 38. Moreover, biochemical parameters, including fasting blood glucose (FBG), C-peptide, triglyceride (TG), total cholesterol (TC), high-density lipoprotein cholesterol (HDL-C), free oxygen radicals testing (FORT), free oxygen radicals defense test (FORD), and serum brain-derived neurotrophic factor (BDNF), were measured in week 42. Finally, rats were humanely sacrificed, and several organs, such as adipose tissues, pancreas, and brains, were collected for further analyses.

### 2.4. Environmental Enrichment (EE) Cages

The EE cages (length: 500 mm × width: 500 mm × height: 300 mm) were equipped with nesting material, a rotating wheel, plastic tubes, and toys (Figure 2). A maximum of three rats can be accommodated in each EE cage. Rats (group DE) were placed in the EE cages for six hours during their dark phase when they were naturally active and had free access to the HFS diet. The EE cages were cleaned, and the EE equipment was changed daily throughout the experimental period [11].

### 2.5. T-Maze Tasks: Spontaneous Alternation and Rewarded Alternation

T-maze apparatus was used to evaluate spatial learning and memory function, and rats were subjected to two tasks, spontaneous alternation and rewarded alternation [12]. The T-shaped apparatus was made up of black acrylic that consisted of a start alley (66 cm L × 16 cm W × 40 cm H) and two arms (50 cm L × 16 cm W × 40 cm H). Guillotine doors were used to close all arms manually (Figure 3). All tasks were carried out in a sound-attenuated room during the dark phase. Rats were adapted to the T-maze environment for two days by placing them in the maze for 30 min daily before commencing any T-maze tasks.

### 2.6. Spontaneous Alternation Task

The spontaneous alternation task was conducted for four days, with six runs performed daily. In each run, a rat was placed at the start arm and allowed to locomote down the start alley and choose either maze arm (all rat’s limbs had to be in the chosen arm). The rat was returned to its home cage for 3 min before the next run. The chosen arms and the total number of alternations made by the rat were recorded. The percentage of spontaneous alternation was calculated as follows: spontaneous alternation % = (total number of alternated arm visits/total number of arm visits) × 100.

### 2.7. Rewarded Alternation Task

This task was performed after the spontaneous alternation task was completed. Food intake was restricted (12 g/rat/day; provided between 8 to 10 a.m.) for two days before the task sessions to maintain high motivation during the T-maze exploration for the food reward. The rewarded alternation task was performed for four days, consisting of six trials daily, with a 3 min inter-trial interval. Each trial had two runs, viz. forced run and choice run. Food reward was placed at the end of the T-maze arms before the trial started. In the forced run, one arm was blocked, and a rat was forced into the open rewarded arm. After the rat ate the food reward, it was returned to its home cage for 3 min before the choice run was carried out. The guillotine door was removed in the choice run, and the rat was free to locomote both arms. If the rat entered the rewarded novel arm, the response was recorded as a correct response. However, if the rat reentered the same arm during the forced run, it was recorded as a wrong response. The percentage of correct responses was calculated as follows: correct response % = (total number of correct responses/total number of trials) × 100.

### 2.8. Measurement of Anthropometric Parameters

The body weight, percentage of weight gain (PWG), body mass index (BMI), and adiposity index (AI) were measured as indicators of obesity in rats. All rats’ body weight was measured every week, whereas PWG, BMI, and AI were measured at the end of the experiment. The total weight gain was determined by subtracting the initial body weight from the final body weight. The percentage of weight gain (PWG) was calculated using the following formula: PWG = total weight gain/initial body weight × 100 [13]. The body mass index (BMI) was calculated as follows: BMI = final body weight/final body length squared (g/cm^2^). The body length was measured between the tip of the nose and the anal region. The BMI for normal adult rats ranged between 0.45 ± 0.02 and 0.68 ± 0.05 g/cm^2^ [14]. The adiposity index (AI) was calculated as follows: AI = (weight of excised white adipose tissues/final body weight) × 100 [15]. The excised white adipose tissue comprised retroperitoneal, epididymal, and mesenteric parts.

### 2.9. Measurement of Fasting Blood Glucose (FBG)

The fasting blood glucose (FBG) samples were collected via the tail vein of the overnight fasted rats and were immediately measured using a glucometer (B/BRAUN omnitest^®^ 3).

### 2.10. Measurement of C-Peptide

Serum C-peptide level was determined using the ELISA kits Elabscience E-EL- R0032. Instead of measuring insulin levels, we measured the serum C-peptide levels for evaluating insulinemia. Similar to insulin, C-peptide is also a good indicator of pancreatic beta-cell function. C-peptide is co-secreted with insulin in equal amounts by pancreatic beta-cells following glucose stimulation [16].

### 2.11. Measurement of the Lipid Profile

Ten microlitres of lateral tail vein blood samples were collected and were instantly analyzed using a handheld CardioChek PA^®^ analyzer (Polymer Technology Systems, Inc., Indianapolis, IN, USA). The triglyceride (TG), total cholesterol (TC), and high-density lipoprotein cholesterol (HDL-C) levels were recorded.

### 2.12. Measurement of Oxidative Status

The systemic oxidative status was assessed by quantitating reactive oxygen species (ROS), antioxidant capacity level, and redox (reductive–oxidative) balance. The ROS was determined using free oxygen radicals testing (FORT). Meanwhile, the free oxygen radicals defense test (FORD) determined the antioxidant capacity level. Both FORT and FORD levels were used to determine the redox balance. These oxidative status analyses were measured using a free oxygen radical analyzer (CR3000 FORM^®^ plus, Callegari, Parma, Italy) according to the manufacturer’s (Callegari, Parma, Italy) instruction [17]. The FORT and FORD results are expressed as mmol H_2_O_2_ Eq/L and mmol/L of Trolox Eq/L, respectively. The reference values of FORT are <2.36 mmol H_2_O_2_ Eq/L with linearity ranging from 1.22 to 4.56 mmol H_2_O_2_ Eq/L. The reference values for FORD are 1.07–1.53 mmol/L of Trolox Eq/L, with the linearity ranging between 0.25 to 3.0 mmol/L of Trolox Eq/L. Redox balance was defined as the ratio of FORD (mmol Trolox Eq/L) to FORT (mmol H_2_O_2_ Eq/L) [18]. Low values of redox balance indicate oxidative stress.

### 2.13. Measurement of Serum BDNF

The serum BDNF was used as an indicator for learning and memory function, as BDNF can cross the blood–brain barrier resulting in a correlation in blood and brain BDNF concentrations [19]. The serum BDNF was determined using the ELISA kits Cusabio Biotech CSB-E04504r.

### 2.14. Tissue Preparation

All rats were euthanized by cervical dislocation at the end of the experiment. The splenic portion of the pancreas, retroperitoneal white adipose tissue (WAT), interscapular brown adipose tissue (BAT), and the brains (right hemispheres) were collected and fixed in 10% neutral buffered formalin for 48 h for further histopathological study. On the other hand, the hippocampi (from the left hemispheres) were immediately removed, placed in cryotubes, and frozen in liquid nitrogen. The hippocampi were stored at −145 °C until gene expression analysis.

### 2.15. Histology

The post-fixed specimens were trimmed accordingly and were individually placed into the labeled histological cassette. The specimens were processed in an automatic tissue processor. Then, the specimens were embedded in a paraffin block and were cut serially at five μm thickness with a rotary microtome. Subsequently, the sections were mounted on microscope slides, and three consecutive sections were stained with haematoxylin and eosin (H&E).

### 2.16. Image Morphometry

Microscopic images were digitally photographed at ×100 and ×200 magnifications using an Olympus BX41 microscope equipped with a microscope camera and imaging software AnalySIS LS Report 2.6. Image processing and morphometric analyses were carried out independently by two blinded observers using computer-aided image analysis software Fiji/ImageJ (Fiji2.3.0) (NIH, Bethesda, MD, USA; http://rsbweb.nih.gov/ij/, accessed on 16 March 2023).

### 2.17. Morphometric Measurement of Adipose Tissue

The retroperitoneal white adipose tissue (WAT) size was measured using the Analyze Particles macro of (Version 1.53t) The average values were obtained from 100 white adipocytes per rat [20]. On the other hand, the percentage of lipid droplet content in the interscapular brown adipose tissue (BAT) was measured using the Measure macro of ImageJ [21]. The average percentage values were obtained from five non-overlapping interscapular sections per rat.

### 2.18. Morphometric Measurement of Pancreatic Islets

The pancreatic islets’ area and circularity index were measured in ten non-overlapping pancreatic fields per rat and were determined using the Analyze Particles macro of ImageJ. The circularity index reports the degree of roundness, where 1.0 corresponds to a perfect circle [22,23].

### 2.19. Morphometric Measurement of Hippocampal Neurons

The hippocampal neurons in CA1 and CA3 regions were quantified in three non-overlapping visual fields per rat. The surviving neuron had a triangular body with a basophilic rim of the cytoplasm. It also has a vesicular nucleus and prominent nucleoli. In contrast, a damaged neuron appeared as a dark and shrunken cell [24]. The percentage of surviving neurons was calculated by dividing the number of surviving neurons by the total number of neurons ×100.

### 2.20. RNA Extraction

Briefly, RNA was isolated from the hippocampal tissues using the RNeasy Mini Kit (catalog no. 74104; Qiagen GmbH, Hilden, Germany). The total RNA concentration was determined by Nanodrop 2000 spectrophotometer (Thermo Fisher Scientific, Waltham, MA, USA). A ratio of absorbance of approximately 2.0 at 260 and 280 nm was acceptable. The extracted total RNA was stored at −80 °C for subsequent use.

### 2.21. Analysis of Gene Expression by the Real-Time PCR

The real-time PCR was performed to evaluate the expression of seventeen genes related to neurogenesis and synaptic plasticity. The analysis was performed in five biological replicates with each technical triplicate (per group).

Following RNA extraction, the total cDNA was synthesized using RT2 First Strand Kit (catalog no. 330401; Qiagen). Next, the cDNA was diluted with RNase-free water and mixed with 2xT2 SYBR Green qPCR Mastermix (catalog no. 330503; Qiagen). This mixture was aliquoted onto the custom RT2 Profiler PCR Array (catalog no. 330171; Qiagen). The custom RT2 Profiler PCR array contained four sets of 21 designed, optimized, and verified RT2 qPCR primer assays, including four reference genes (*Actb*, *Pgk1*, *Hprt1*, and *Tbp*), a proprietary control panel to monitor genomic DNA contamination (GDC), the first strand synthesis (RTC) and real-time PCR efficiency (PPC) (Table 1). The cDNA was used as the PCR template and was amplified using the Applied Biosystems Step One Plus™ Real-Time PCR system (Life Technologies™ Waltham, MA, USA) under the following thermal cycling conditions: 10 min at 95 °C, followed by 40 cycles of 15 s at 95 °C and 1 min at 60 °C.

The generated cycle threshold (CT) values from each reaction were exported to the Expression Suite version 1.3 (Life Technologies/Applied Biosystems, Waltham, MA, USA) to rapidly and accurately quantify relative gene expression across many genes and samples. The relative gene expression was calculated using the 2^−ΔΔCt^ method [25]. Four reference genes (*Actb*, *Pgk1*, *Hprt1*, and *Tbp*) were used to normalize the reactions, and the C and D groups were used as calibrators. Changes in gene expression were analyzed with the Student’s *t*-test followed by Benjamini Hochberg correction.

The fold-change values were expressed LOG2 fold-changes (calibrator groups set to 0). The fold changes lower than 1 (decreased gene expression) were converted to a negative number, and fold changes greater than 1 (increased gene expression) were converted to a positive number. Genes were identified as differentially expressed if FDR q <  0.05 and LOG2 fold-change of value > 1.0-fold.

### 2.22. Statistical Analysis

Statistical analysis for the lipid panel and morphometric analyses were performed using the non-parametric Kruskal–Wallis test, followed by Dunn’s multiple comparison test. Data were reported as medians with interquartile ranges. Other parameters were measured using the one-way analysis of variance (ANOVA) with the Bonferroni post-hoc test and were reported as mean ± standard deviation (SD). All statistical analyses were carried out using a GraphPad Prism version 8.0 (GraphPad Software, Inc, San Diego, CA, USA) except for the gene expression analysis, which was performed using Expression Suite software described in the methodology. *p* of <0.05 was considered statistically significant.

## 3. Results

There were no significant differences in the baseline parameters between all rats.

### 3.1. Effects of EE Exposure and Metformin Treatment on Fasting Blood Glucose (FBG), Serum C-Peptide, Histological Features, and Morphometric Measurement of Pancreatic Islets in HFS Diet-Induced T2D Rats

Previous studies proved the association between overconsumption of the HFS diet and T2D. Typically, T2D is characterized by hyperglycemia and hyperinsulinemia. To investigate the possible anti-diabetic effects of EE and metformin interventions in HFS diet-induced T2D rats, we measured two T2D indicators, FBG, and serum C-peptide.

As expected, the D group had higher FBG and serum C-peptide levels than the C group (*p* < 0.05), whereas the DM group had lower FBG and serum C-peptide levels than the D group (*p* < 0.05). Surprisingly, the DE group had a similar outcome to the DM group (*p* < 0.05). (Table 2). This critical finding implied that EE exposure triggered anti-hyperglycemic and anti-hyperinsulinemic effects in HFS diet-induced T2D rats.

Next, we determined the effects of EE and metformin interventions on the histology and morphometry of pancreatic islets in HFS diet-induced T2D rats. Our result (Figure 4) showed that the D group had oversized and irregular islets. Furthermore, some parts of the islets in the D group exhibited vacuolated cells with pyknotic nuclei. The EE and metformin interventions remarkably prompted islet recovery. The DE and DM groups had smaller and partially irregular islets than the D group. Although these islets were moderately rescued, vacuolated cells were almost undetected.

The morphometric analysis significantly validated these histological observations (Figure 4). The mean area confirmed the size of the pancreatic islets. In contrast, the mean circularity confirmed the shape of the pancreatic islets, respectively. These results indicated that EE and metformin interventions promoted recovery and preserved pancreatic islets in HFS diet-induced T2D rats.

### 3.2. Effects of EE Exposure and Metformin Treatment on Serum Lipid Profile, Anthropometric Parameters, Histological Features, and Morphometric Measurement of Adipose Tissue in HFS Diet-Induced T2D Rats

Other than T2D, the HFS diet is also induced dyslipidemia and obesity. Dyslipidemia is characterized by an odd combination of high TG and TC levels and low HDL-C levels [26]. Hence, we measured the serum lipid profile to investigate the possible anti-dyslipidemic effects of EE and metformin interventions in HFS diet-induced T2D rats.

By the end of this study, we found that the D group had higher levels of TG and TC and lower levels of HDL-C than the control (*p* < 0.05). Meanwhile, DE and DM groups had lower levels of TG and TC than the D group (*p* < 0.05), but their HDL-C levels were insignificant (Table 3). These results indicated that the interventions of EE and metformin promoted the anti-dyslipidemic activities in HFS diet-induced T2D rats.

Furthermore, we measured the anthropometric parameters to investigate the possible anti-obesity effects of EE and metformin interventions in HFS diet-induced T2D rats. Based on Table 3, the anthropometric parameters, including final body weight, PWG, and BMI of the D group, were higher than the C group (*p* < 0.05). On the contrary, both DE and DM groups had lower anthropometric parameters than the D group (*p* < 0.05). This finding implied that the interventions of EE and metformin promoted bodyweight reduction in HFS diet-induced T2D rats.

Next, we determined whether the body weight reduction was due to adipose tissue loss. For this purpose, we analyzed the AI, histological features, and morphometric measurement of adipose tissue. Our results showed that the D group’s AI was higher than the C group’s (*p* < 0.05), while both DE and DM groups had lower AI than the D group (*p* < 0.05) (Table 4). Moreover, the histological analysis revealed that the WAT in the D group was more extensive than in the C group. In contrast, the WAT in DE and DM groups was less extensive than in the D group. As for the lipid droplets in BAT, the D group had larger droplets than the C group. Meanwhile, DE and DM groups had smaller droplets than the D group (Figure 5). Furthermore, the results of the morphometric analysis confirmed these observations (Table 5). These outcomes indicated that EE and metformin interventions in HFS diet-induced T2D rats promoted adipose tissue loss.

### 3.3. Effects of EE Exposure and Metformin Treatment on Oxidative Status in HFS Diet-Induced T2D Rats

Overconsumption of the HFS diet leads to T2D, which is also linked to the increment of oxidative stress [27]. This study also determined whether EE and metformin interventions have the anti-oxidative capacity to overcome the oxidative stress induced by HFS diet-induced T2D. It was performed by assessing serum FORT, FORD, and redox balance levels.

Our results (Table 4) revealed that the serum FORT level was higher in the D group than in the C group (*p* < 0.05) and exceeded the threshold of 2.36 mmol H_2_O_2_ Eq/L. It indicated that the HFS diet initiated the reactive oxygen species (ROS) and increased oxidative stress. Although the serum FORT levels in DE and DM groups exceeded the threshold, they were significantly lower than in the D group. Nevertheless, all groups had normal serum FORD levels, indicating that all groups had normal antioxidant levels. The FORT and FORD levels were used to calculate the redox balance ratio. As a result, the D group had a lower redox balance ratio than the C group (*p* < 0.05), indicating the oxidative stress state. More importantly, the redox balance ratio of the DE (*p* < 0.05) and DM (*p* > 0.05) groups was higher than the D group. Overall, these results indicated that EE and metformin interventions alleviated oxidative stress in HFS diet-induced T2D rats.

### 3.4. Effects of EE Exposure and Metformin Treatment on Spatial Learning and Memory Function, Serum BDNF, Histological Features, and Morphometric Measurement of Hippocampal Neurons in HFS Diet-Induced T2D Rats

A recent study has proved that T2D is significantly associated with hippocampal neuronal death and hippocampal-dependent memory impairments [28]. Herein, we investigated whether EE and metformin interventions improve hippocampal neuron survival and hippocampal-dependent memory function in HFS diet-induced T2D rats. For this purpose, we evaluated the T-maze tasks, serum BDNF levels, histology, and morphometry of hippocampal neurons.

In this study, the T-maze tasks (spontaneous alternation and rewarded alternation) were performed to evaluate the effects of EE and metformin interventions on spatial learning and memory function in HFS diet-induced T2D rats. Table 5 shows that the D group had lower percentages of correct response and spontaneous alternation than the C group (*p* < 0.05). Contrarily, DE and DM groups had a higher correct response and spontaneous alternation percentages than the D group (*p* < 0.05). These results indicated that diabetes caused spatial learning and memory deficits, but more importantly, the EE and metformin interventions significantly ameliorated these deficits in HFS diet-induced T2D rats.

Next, we investigated the serum BDNF levels in all rats. We found that the D group’s serum BDNF levels were lower than the C group (*p* < 0.05). On the contrary, DE and DM groups had higher serum BDNF levels than the D group (*p* < 0.05) (Table 5). The increment of the serum BDNF in HFS diet-induced T2D rats that received either EE or metformin indicated that these interventions improved hippocampal neuron survival and hippocampal-dependent memory function.

As learning and memory relate to the hippocampus, we evaluated the histological features of the pyramidal neurons in the hippocampus subregions CA1 and CA3 that are important in memory formation. Many CA1 and CA3 pyramidal neurons in the D group were dark-stained and shrunken. On the other hand, most of the pyramidal neurons (in both subregions) in the DE and DM groups were healthy and appeared as triangular bodies. They had basophilic rims of the cytoplasm, vesicular nuclei, and prominent nucleoli (Figure 6).

The results of the morphometric analysis confirmed these histological findings. The D group had less surviving neuron percentage in CA1 and CA3 subregions than the C group (*p* < 0.05). In contrast, DE and DM groups had more surviving neuron percentages than the D group (*p* < 0.05) (Figure 6). These outcomes indicated that HFS diet-induced T2D triggered hippocampal neuronal death. More importantly, EE and metformin interventions enhanced hippocampal neuron survival in HFS diet-induced T2D rats.

### 3.5. Effects of EE Exposure and Metformin Treatment on Gene Expression in the Hippocampus of HFS Diet-Induced T2D Rats

In addition to mentioned parameters, we also investigated the expression of the hippocampal genes in all rats. The real-time PCR was performed to evaluate the expression of seventeen hippocampal genes related to neurogenesis and synaptic plasticity. We first investigated the expression of candidate genes of group D relative to the C group. Our results revealed that the expressions of *Bdnf*, *Irs1*, *Pik3ca*, *Bcl2*, and *Atf4* were significantly decreased (downregulated) in the D group compared to the C group. In contrast, the expression of *Gsk3b* was significantly increased (upregulated) in the D group compared to the C group (Figure 7). These results suggested that HFS diet-induced T2D impairs neurogenesis mediated through the altered expression of these genes.

For the last part of this study, we investigated the expression of candidate genes of group DE and DM groups relative to the D group. Interestingly, we found that the expressions of *Bdnf*, *Ntrk2*, *Irs1*, *Bcl2*, *Atf4*, *Map2k1*, *Mapk1*, *Camk2g*, *Grin1*, and *Gria2* were significantly increased (upregulated) in the DE group compared to the D group. Meanwhile, the expression of *Gsk3b* was significantly decreased (downregulated) in the DE group compared to the D group (Figure 8). These results suggested that EE exposure promotes neurogenesis and synaptic plasticity by altering the expression of these genes in HFS diet-induced T2D rats. In addition, the metformin treatment also promotes neurogenesis and synaptic plasticity by altering the expression of *Ntrk2*, *Irs1*, *Gsk3b*, *Bcl2*, *Map2k1*, and *Grin1* genes in HFS diet-induced T2D rats.

## 4. Discussion

Extensive studies revealed the correlation between a Western-style diet with obesity and T2D. Eventually, individuals with T2D may develop neurodegeneration and memory impairment. Such impairments are due to direct or indirect T2D consequences such as hyperglycemia, insulin resistance, adipose tissue dysfunction, and oxidative stress [29,30,31]. Therefore, it is crucial to identify effective therapeutic strategies for these impairments, and the medicine-free approach remains a priority.

In general, this study provides rodent model evidence that EE and metformin interventions improved hippocampal neuron survival and hippocampal-dependent memory function in HFS diet-induced T2D. In addition, our study also shows that EE has similar anti-diabetic properties to metformin by alleviating metabolic dysfunctions in HFS diet-induced T2D rats.

Environmental enrichment (EE) with ample space and various stimulators provides physical exercise opportunities, entertaining activities, learning experiences, and complex social interactions [5]. Studies proved that EE promotes hippocampal neurogenesis and memory function. In addition, EE also enhances the expression of neurotrophic factors [7]. Previous studies also suggested that exercise (such as in EE) is beneficial in treating the metabolic dysfunctions of the T2D [32,33]. In the following paragraphs, we will discuss the findings of this study, the similar anti-diabetic effects of EE to metformin, and, more importantly, the neuroprotective effects of the individual intervention of EE and metformin.

The benefit of metformin use in T2D is well acknowledged. Still, many studies explored alternative strategies for treating T2D and its complications. In our study, the medicine-free EE shows similar anti-diabetic medicinal effects to metformin, including anti-hyperglycemia, anti-hyperinsulinemia, anti-dyslipidemia, and anti-oxidative stress. In addition, EE and metformin interventions also preserved pancreatic islets and promoted adipose tissue loss in our rat model of HFS diet-induced T2D.

In our study, the EE exposure reduced the serum glucose in the HFS diet-induced T2D rats. We believe that the exercise component of EE plays a crucial strategy in glycemic control. Exercise promotes energy expenditure and induces muscular contraction-mediated glucose uptake, which facilitates the removal of excess glucose from blood circulation (hypoglycemic effect) [34]. Our findings also showed that EE exposure alleviated the oversecretion of insulin in the HFS diet-induced T2D rats. Although we measured the serum C-peptide in our study, its level is proportionate to the insulin level. The C-peptide is part of proinsulin, cleaved prior to co-secretion with insulin by the pancreatic beta-cells [16]. The histology and morphometry of pancreatic islets were also investigated, and our study showed that the EE exposure preserved the structure of the pancreatic islets in the HFS diet-induced T2D rats. We assume that the decreased serum C-peptide and the preserved pancreatic islet morphology were possibly due to the alleviation of the glycemic control in the HFS diet-induced T2D rats. It eventually inhibits the overstimulation of pancreatic islets and the over secretion of the insulin [34].

Our study also showed that EE exposure promoted weight loss and alleviated BMI, AI, and dyslipidemia in the HFS diet-induced T2D rats. Again, we believe that exercise in EE plays a vital role in these changes. Although our EE setting lacks intense resistance or endurance exercises, we believed the exercise was sufficient to promote energy expenditure and enhance the lipolysis [35]. The exercise generates a negative energy balance that triggers body weight loss and improves BMI [36]. The EE mechanisms responsible for alleviating dyslipidemia and AI may relate to the increased lipoprotein lipase activity that mediates triglyceride hydrolysis and hepatic cholesterol excretion [37]. The histology and morphometry of WAT and BAT confirmed these findings. The exercise alleviates WAT and BAT in the HFS diet-induced T2D rats. It may involve the elevation of mitochondrial activity, adipokine secretion alteration, glucose uptake, and lipidome reductions [38,39].

Identical to metformin, the EE exposure also reduced the oxidative stress in the HFS diet-induced T2D rats. However, the underlying mechanism of the anti-oxidative activity of EE and metformin is not fully understood. Few studies revealed that metformin increased the expression of the nuclear factor E2-related factor 2 (Nrf2) protein. The increment of Nrf2 protein activated antioxidant defense systems such as superoxide dismutase (SOD) [9]. Meanwhile, a recent study proposed that EE can eliminate oxidative stress by altering the detoxifying metabolism. The detoxifying metabolism has three phases that modify toxins into active metabolites, catalyze the active metabolites into hydrophilic products with transferase enzymes, and excrete the final products via various transporter systems. The EE regulates enzymes participating in the detoxification metabolism, such as cytochrome P450 family 1 subfamily A member 2 (Cyp1a2) and carbonyl reductase2 (Cbr2) [40]. The anti-oxidative activities of EE and metformin might be responsible for anti-diabetic and neuroprotective effects in the HFS diet-induced T2D rats.

The main findings of this study indicated that individual intervention of EE and metformin improved the serum BDNF, hippocampal-dependent memory function (T-maze tasks), and hippocampal neuron survival in the HFS diet-induced T2D rats. In conjunction with these pieces of evidence, the individual intervention of EE and metformin also improved the expression of hippocampal genes related to neurogenesis and synaptic plasticity in the HFS diet-induced T2D rats.

T-maze is an animal-friendly procedure and a valuable tool for validating spatial learning and memory functions. This procedure is based on the natural tendency of animals to explore and memorize their environment to obtain food. The present study showed that the individual intervention of EE and metformin increased the correct response and spontaneous alternation percentages. These results indicated that these interventions improved hippocampal-dependent memory functions in the HFS diet-induced T2D rats. Brain-derived neurotrophic factor (BDNF) is one of the possible mediators responsible for improving these rats’ learning and memory function [19].

The BDNF is a vital neurotrophin involved in neuronal development, survival, maintenance, learning, and memory function. It is widely expressed throughout the central nervous system and released into the blood circulation by the brain. The availability of the serum BDNF is another piece of evidence that indicates the hippocampal learning and memory functions are preserved [41]. In our study, the individual intervention of EE and metformin significantly increased the serum BDNF in the HFS diet-induced T2D rats. We proposed that the possible mechanisms of EE and metformin involved the alteration of BDNF secretion and expression.

Hippocampus is a brain structure with several critical subregions, the cornu ammonis (CA: CA1, CA2, and CA3) and the dentate gyrus. These regions are composed of pyramidal neurons that support cognitive function, including but not limited to learning and memory response [42]. Our results revealed that the individual intervention of EE and metformin significantly enhanced hippocampal neuron survival in the HFS diet-induced T2D rats. We found that the T2D rats that received either the EE or metformin interventions had a higher percentage of surviving neurons (in the CA1 and CA3 subregions) than the non-treated T2D rats. We suggested that EE and metformin’s anti-diabetic and anti-oxidative stress activities are the possible mechanisms involved in hippocampal neuron survival enhancement.

We also proposed that the EE and metformin promote the expression of hippocampal genes related to neurogenesis and synaptic plasticity in the HFS diet-induced T2D rats. These genes regulate the binding of BDNF/TrkB and its downstream signaling, including PI3K-Akt and Ras–MAPK pathways. The BDNF binds to tyrosine receptor kinase B (TrkB) and activates three main intracellular signaling cascades. The BDNF/TrkB-signalling promotes neuron survival by activating the phosphatidylinositol 3 kinase pathway (PI3K-Akt) and cellular proliferation by the Ras–mitogen-activated protein kinase pathway (Ras–MAPK). Moreover, BDNF also promotes memory formation by regulating the phospholipase Cγ pathway (PLCγ–Ca^2+^) [43]. Thus, we assessed the hippocampal expression of seventeen hippocampal genes involved in these signaling pathways to investigate the mechanisms responsible for these improvements.

The *Bdnf*, *Irs1*, *Pik3ca*, *Atf4*, and *Bcl2* were downregulated, and the *Gsk3b* was upregulated in the HFS diet-induced T2D rats. The altered gene expressions may cause impairments of neurogenesis, synaptic plasticity, and memory formation in these rats. The downregulation of *Bdnf*, the gene that encodes the neurotrophin BDNF, directly disrupts the BDNF/TrkB binding, PI3K-Akt, Ras–MAPK, and PLCγ–Ca^2+^ pathways [44]. In particular, the downregulation of *Irs1*, the gene that encodes the insulin receptor substrate-1 (IRS-1), inhibits the signaling cascade of PI3K-Akt [45]. Other than cognitive dysfunction, the abnormal expression of IRS1 is also related to T2D and hippocampal insulin resistance [46]. Meanwhile, the downregulated *Pik3ca* disrupts the availability of phosphoinositide-3-kinase, which interferes with the binding and activation of protein kinase Akt. This interference leads to the deactivation of anti-apoptotic cytokines, thereby impeding the hippocampal neuron survival [45]. The *Gsk3b* upregulation promotes pro-apoptotic proteins, whereas the Bcl2 downregulation inhibits anti-apoptotic proteins. The alteration of the expression of these genes decreases hippocampal neuron survival (neurogenesis) [47]. The *Atf4* gene encodes cyclic AMP-responsive element-binding protein (CREB) that mediates the transcription of essential prosurvival genes and other downstream targets involved in neural plasticity. Thus, the downregulation of the *Atf4* promotes neuron death, glutamatergic synaptic dysfunction, and memory deficits [48].

The EE exposure given to the HFS diet-induced T2D rats resulted in the upregulation of *Bdnf*, *Ntrk2*, *Irs1*, *Bcl2*, *Atf4*, *Map2k1*, *Mapk1*, *Camk2g*, *Grin1*, and *Gria2*, and the downregulation of *Gsk3b*. The *Ntrk2* encodes the tyrosine kinase receptor, TrkB, a neurotrophin receptor activated by BDNF. Thus, the upregulations of *Bdnf* and *Ntrk2* improve the binding of BDNF/TrkB and its downstream signaling [43]. Next, the upregulation of *Irs1* promotes the activation of the PI3K-Akt signaling cascade. This finding also suggested that EE exposure may ameliorate insulin resistance in the hippocampus. The EE exposure also resulted in the downregulation of *Gsk3b* and the upregulation of *Bcl2*. These alterations inhibit the pro-apoptotic proteins and promote anti-apoptotic proteins, which enhance hippocampal neuron survival. The *Map2k1* encodes the dual specificity mitogen-activated protein kinase kinase 1, and the *Mapk1* encodes the mitogen-activated protein kinase 1. Both kinases are essential components of the Ras–MAPK pathway. Therefore, the upregulations of *Map2k1* and *Mapk1* indicate that EE exposure promotes hippocampal neuron survival. The *Camk2g* encodes the calcium/calmodulin-dependent protein kinase type II subunit gamma, one of the components of the PLCγ–Ca^2+^ pathway. Hence, the upregulation of *Camk2g* promotes synaptic plasticity, long-term potentiation (LTP), and memory formation. *Grin1* encodes the component of NMDA glutamate receptor complexes, whereas *Gria2* encodes the AMPA glutamate receptor. These two glutamate receptors mediate LTP at CA3–CA1 synapses. Therefore, the upregulations of *Grin1* and *Gria2* promote long-term potentiation (LTP) and memory formation.

Our study also indicated that the metformin treatment successfully enhances hippocampal neuron survival and promotes hippocampal-dependent learning and memory in the HFS diet-induced T2D rats by altering the expression of *Ntrk2*, *Irs1*, *Gsk3b*, *Bcl2*, *Map2k1*, and *Grin1* genes. Previous studies revealed that metformin might help inhibit neuronal cell death through the p53 signal-induced apoptosis [9,49]. However, our study revealed that metformin improves neuron survival by altering the expression of genes related to the BDNF/TrkB and its downstream signaling, including PI3K-Akt and Ras–MAPK pathways. Our study also showed that metformin promotes hippocampal-dependent memory by altering genes related to LTP (*Grin1*).

## 5. Conclusions

In summary, our study revealed that the overconsumption of the Western-style diet instigated T2D in rats. These rats had hyperglycemia, hyperinsulinemia, dyslipidemia, and histological alterations in pancreatic islets and adipose tissue. These rats also had high levels of systemic oxidative stress. Nevertheless, the EE had similar anti-diabetic properties to metformin by alleviating metabolic dysfunctions and histological features in the HFS diet-induced T2D rats. Like metformin, the EE also reduced the systemic oxidative stress in the HFS diet-induced T2D rats.

Chronically, hippocampal neuronal death and hippocampal-dependent memory impairment were triggered in the HFS diet-induced T2D rats. These rats had low percentages of T-maze tasks, low serum BDNF and low percentages of hippocampal neuron survival in CA1 and CA3. We also found that the expressions of hippocampal genes related to neuron survival were altered in these rats.

The highlight of this study is the investigation of the neuroprotective effects of EE and metformin in HFS-induced T2D rats with hippocampal neuron and memory impairments. The interventions of EE and metformin significantly improved hippocampal neuron survival and hippocampal-dependent memory functions in the HFS-induced T2D rats. Our findings revealed that the HFS diet-induced T2D rats either exposed to EE or treated with metformin had high percentages of T-maze tasks, high serum BDNF and high percentages of hippocampal neuron survival in CA1 and CA3. Although the underlying mechanisms of EE and metformin were not fully understood, we believed that the anti-oxidative activities of these two interventions might be responsible for both anti-diabetic and neuroprotective effects in the HFS diet-induced T2D rats. Specifically, we proposed that the underlying mechanisms of these interventions involved the expression changes of hippocampal genes related to neurogenesis and synaptic plasticity. The EE exposure improved the expressions of genes that regulate the BDNF/TrkB binding and its downstream signaling, including PI3K-Akt, Ras–MAPK, and PLCγ–Ca^2+^ pathways. Our study also showed that EE enhanced the expression of genes related to LTP. Meanwhile, the metformin treatment only improved the expressions of genes that regulate the BDNF/TrkB binding, PI3K-Akt, Ras–MAPK, and LTP.

In conclusion, environmental enrichment (EE) and metformin improved metabolic functions, hippocampal neuron survival, and hippocampal-dependent memory in HFS diet-induced T2D rats. The underlying neuroprotective mechanisms of these interventions involved the expression of genes that regulate the BDNF/TrkB binding, PI3K-Akt, Ras–MAPK, PLCγ–Ca^2+^, or LTP. The details of these mechanisms should be further verified in future studies.

## Figures and Tables

**Figure 1 biology-12-00480-f001:**
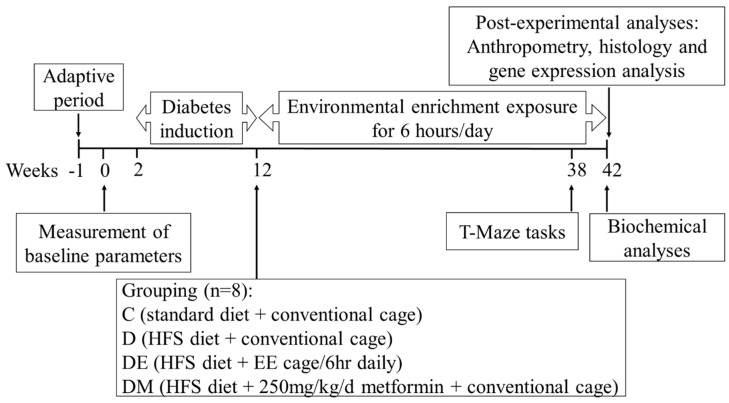
The timeline of experimental design. Following the adaptive week, baseline parameters (body weight, BMI, biochemical parameters, and T-maze tasks) were measured. During the diabetes induction period, rats in the control group (C group, *n* = 8) were placed in conventional polypropylene cages. They were maintained on an ad libitum standard chow pellet diet and tap water. In contrast, the remaining rats (*n* = 24) were fed the HFS diet to induce T2D. Rats with FBG above 6.7 mmol/L were randomly subdivided into three groups of eight rats (D, DE, and DM groups) and were continuously fed the HFS diet until the end of the study. As interventions, rats in the DE group were placed in EE cages for 6 h/day, and rats in the DM group were given metformin once daily. Bodyweight measurement was taken every week. The T-maze tasks were performed again in week 38. Moreover, biochemical parameters, including FBG, C-peptide, TG, TC, HDL-C, FORT, FORD, and serum BDNF were measured in week 42. At the end of the experiment, rats were humanely sacrificed. Several organs were collected for post-experimental analyses. Abbreviations: BDNF, serum brain-derived neurotrophic factor; BMI, body mass index; EE, environmental enrichment; FBG, fasting blood glucose; FORD, free oxygen radicals defense test; FORT, free oxygen radicals testing; HDL-C, high-density lipoprotein cholesterol; HFS, high-fat diet plus sucrose solution drink; TC, total cholesterol; TG, triglyceride; T2D, type 2 diabetes.

**Figure 2 biology-12-00480-f002:**
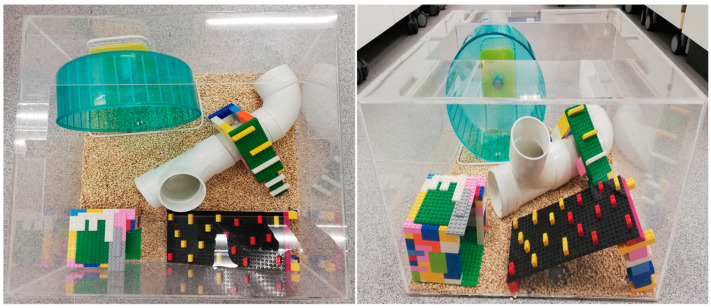
Top view (**left panel**) and side view (**right panel**) of the environmental enrichment (EE) cage. The EE cage was equipped with nesting material, a rotating wheel, a plastic tube, and toys.

**Figure 3 biology-12-00480-f003:**
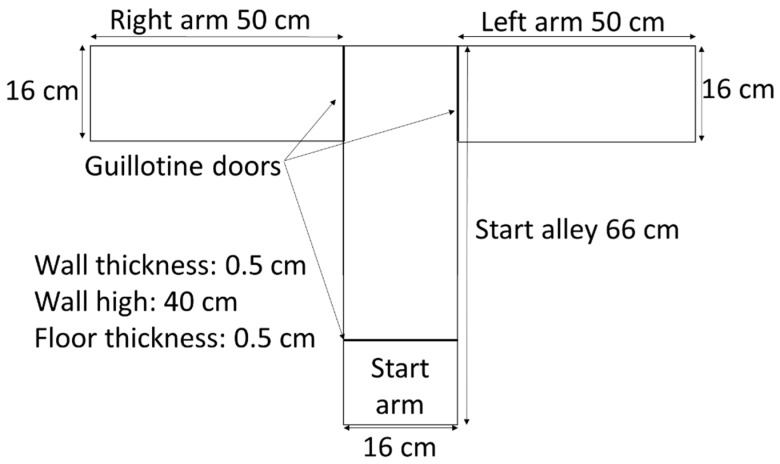
T-maze design. The T-shaped platform apparatus was made up of black acrylic with a start alley (66 cm L × 16 cm W × 40 cm H) and two arms (50 cm L × 16 cm W × 40 cm H). Guillotine doors were used to close all arms. The thickness of the walls and the floor was 0.5 cm.

**Figure 4 biology-12-00480-f004:**
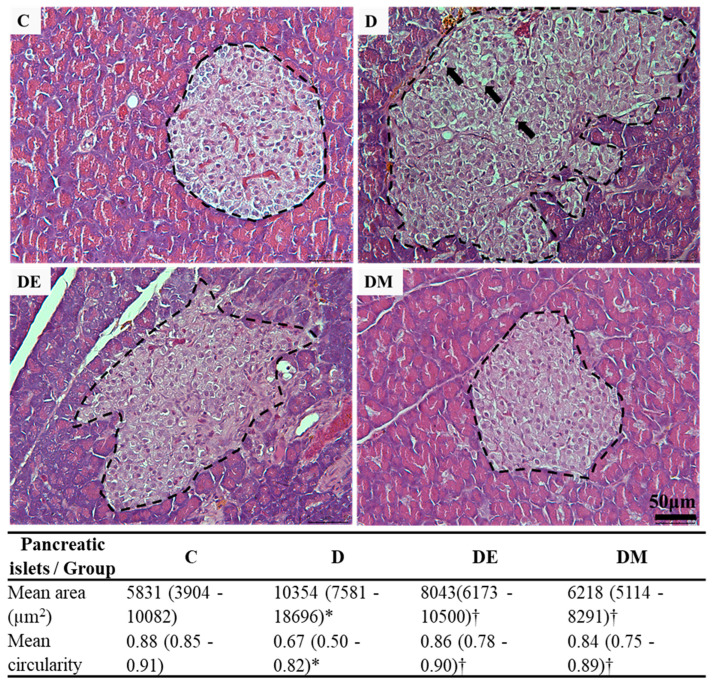
Effects of EE exposure and metformin treatment on histological features and morphometric measurement of pancreatic islets in HFS diet-induced T2D rats. The dashed line traces the pancreatic islets in these photomicrographs. The D group had more oversized and irregular islets than the C group. In contrast, DE and DM groups had smaller and partially irregular islets than the D group. The islets of the D group consisted of vacuolated cells with pyknotic nuclei (arrows). The vacuolated cells were almost undetected in DE and DM groups (H&E, Scale bar: 50 μm). The results of the morphometric analysis confirmed these observations. The mean area and the mean circularity confirmed the size and shape of the pancreatic islets, respectively. The morphometric data are expressed as median (percentile 25–75) (non-parametric k-independent median test). Values showing * *p* < 0.05 vs. C; † *p* < 0.05 vs. D. Abbreviations: C, control rats fed standard chow pellet diet and water; D, HFS diet-induced T2D rats; DE, HFS diet-induced T2D rats exposed to EE; DM, HFS diet-induced T2D rats treated with metformin.

**Figure 5 biology-12-00480-f005:**
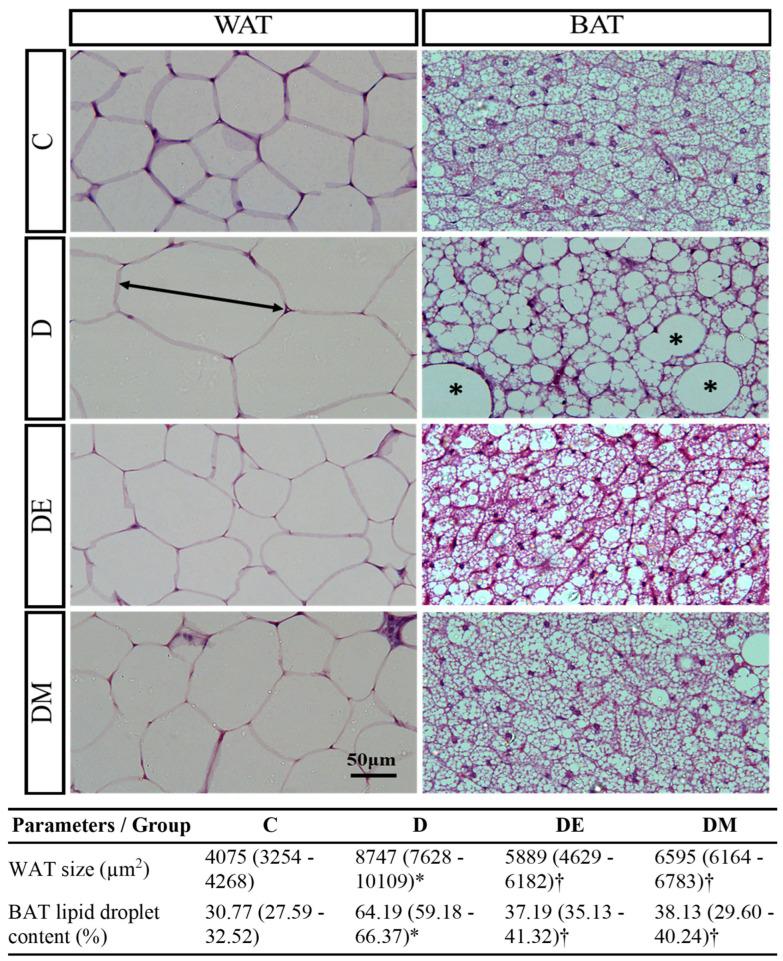
Effects of EE exposure and metformin treatment on histological features and morphometric measurement of WAT and BAT in HFS diet-induced T2D rats. The D group had extensive WAT size (double arrow line) and more lipid droplets in BAT (asterisks) than DE and DM groups (H&E, Scale bar: 50 μm). The results of the morphometric analysis confirmed these observations. The morphometric data are expressed as median (percentile 25–75) (non-parametric k-independent median test). Values showing * *p* < 0.05 vs. C; † *p* < 0.05 vs. D. Abbreviations: C, control rats fed standard chow pellet diet and water; D, HFS diet-induced T2D rats; DE, HFS diet-induced T2D rats exposed to EE; DM, HFS diet-induced T2D rats treated with metformin; BAT, brown adipose tissue; WAT, white adipose tissue.

**Figure 6 biology-12-00480-f006:**
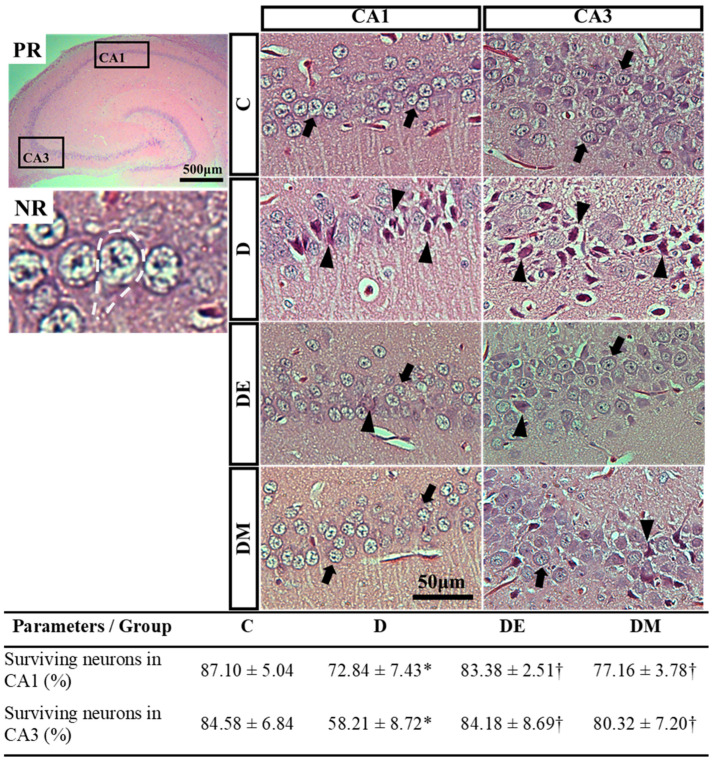
Effects of EE exposure and metformin treatment on histological features and morphometric measurement of hippocampal neurons in HFS diet-induced T2D rats. PR: Pictorial representation (PR) of hippocampus highlighting the investigated CA1 and CA3 subregions. NR: Neuronal representation showing the morphology of typical neurons (white dashed line). Neuronal morphology was evaluated via haematoxylin and eosin (H&E) staining. The D group had more damaged neurons (triangle) in both subregions than the C group. In contrast, DE and DM groups had more surviving neurons (arrow) than the D group. The results of the morphometric analysis confirmed these observations. The morphometric data are expressed as mean ± standard deviation (one-way ANOVA; post hoc Bonferroni). Values showing * *p* < 0.05 vs. C; † *p* < 0.05 vs. D. Abbreviations: C, control rats fed standard chow pellet diet and water; D, HFS diet-induced T2D rats; DE, HFS diet-induced T2D rats exposed to EE; DM, HFS diet-induced T2D rats treated with metformin; CA, cornu ammonis.

**Figure 7 biology-12-00480-f007:**
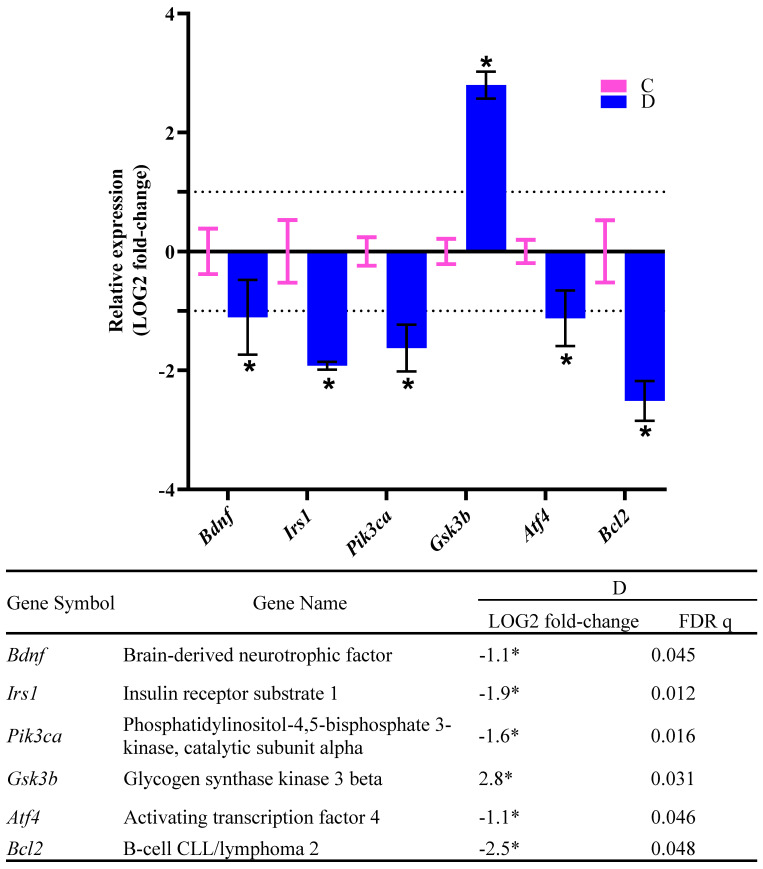
Effects of HFS diet-induced T2D on the expression of genes related to neurogenesis and synaptic plasticity in rats. The graph and table show the relative expression of genes of the D group relative to the C group. The columns and error bars indicate the mean and standard deviation of LOG2 fold-change values in the C (pink) and D (blue) groups. The LOG2 fold-change values of the C group were set to zero. The asterisks indicate LOG2 fold-change >1.0 with FDR q < 0.05. Dotted lines indicate the LOG2 fold-change threshold values.

**Figure 8 biology-12-00480-f008:**
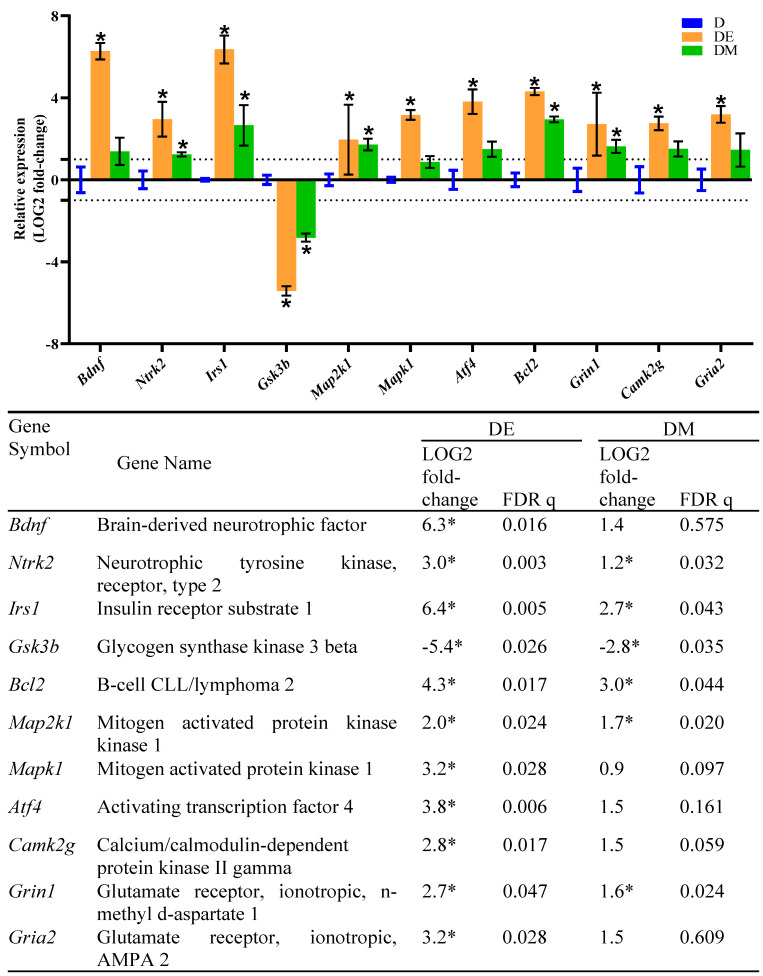
Effects of EE exposure and metformin treatment on the expression of genes related to neurogenesis and synaptic plasticity in HFS diet-induced T2D rats. The graph and table show the relative expression of genes of the DE and DM groups relative to the D group. The columns and error bars indicate the mean and standard deviation of LOG2 fold-change values in the D (blue), DE (orange) and DM (green) groups. The LOG2 fold-change values of the D group were set to zero. The asterisks indicate LOG2 fold-change >1.0 with FDR q < 0.05. Dotted lines indicate the LOG2 fold-change threshold values.

**Table 1 biology-12-00480-t001:** List of inclusion genes for the Custom RT2 Profiler PCR Array.

RT2 qPCR Primer Assay Catalogue No.	Gene	NCBI Reference Sequence:	Official Name
PPR45425C	*Akt1*	NM_033230	v-akt murine thymoma viral oncogene homolog 1
PPR42749A	*Atf4*	NM_024403	activating transcription factor 4
PPR06577B	*Bcl2*	NM_016993	B-cell CLL/lymphoma 2
PPR45333A	*Bdnf*	NM_012513	Brain-derived neurotrophic factor
PPR43356B	*Calm1*	NM_001007614	calmodulin 1
PPR45142B	*Camk2g*	NM_133605	calcium/calmodulin-dependent protein kinase II gamma
PPR43063B	*Grb2*	NM_030846	growth factor receptor-bound protein 2
PPR06753B	*Gria2*	NM_001083811	glutamate receptor, ionotropic, AMPA 2
PPR06847D	*Grin1*	NM_017010	glutamate receptor, ionotropic, N-methyl D-aspartate 1
PPR44848A	*Gsk3b*	NM_032080	glycogen synthase kinase 3 beta
PPR44872B	*Irs1*	NM_012969	insulin receptor substrate 1
PPR47860F	*Kras*	NM_031515	KRAS proto-oncogene, GTPase
PPR43465A	*Map2k1*	NM_031643	mitogen-activated protein kinase 1
PPR48780A	*Mapk1*	NM_053842	mitogen-activated protein kinase 1
PPR45322B	*Ntrk2*	NM_001163168	neurotrophic tyrosine kinase, receptor, type 2
PPR59984F	*Pik3ca*	NM_133399	phosphatidylinositol-4,5-bisphosphate 3-kinase, catalytic subunit alpha
PPR45212A	*Sh2b1*	NM_001048180	SH2B adaptor protein 1
PPR06570C	*Actb*	NM_031144	actin, beta
PPR42247F	*Hprt1*	NM_012583	hypoxanthine phosphoribosyltransferase 1
PPR56649C	*Pgk1*	NM_053291	phosphoglycerate kinase 1
PPR47412A	*Tbp*	NM_001004198	TATA box binding protein
PPR63338A	GDC		Genomic DNA control
PPX63339A	PPC		PCR positive control
PPX63340A	RTC		Reverse transcription control

**Table 2 biology-12-00480-t002:** Effects of EE exposure and metformin treatment on fasting blood glucose and serum C-peptide in HFS diet-induced T2D rats.

Parameters/Group	C	D	DE	DM
FBG (mmol/L)	5.90 ± 0.16	8.26 ± 0.48 *	6.76 ± 0.10 ^†^	6.76 ± 0.34 ^†^
C-peptide (ng/mL)	26.19 ± 13.65	94.30 ± 26.26 *	53.03 ± 33.31 ^†^	40.35 ± 22.15 ^†^

Data are expressed as mean ± standard deviation (one-way ANOVA; post hoc Bonferroni). Values showing * *p* < 0.05 vs. C; ^†^
*p* < 0.05 vs. D. Abbreviations: C, control rats fed standard chow pellet diet and water; D, HFS diet-induced T2D rats; DE, HFS diet-induced T2D rats exposed to EE; DM, HFS diet-induced T2D rats treated with metformin; FBG, fasting blood glucose.

**Table 3 biology-12-00480-t003:** Effects of EE exposure and metformin treatment on serum lipid profile and anthropometric parameters in HFS diet-induced T2D rats.

Parameters/Group	C	D	DE	DM
TG (mmol/L)	0.98 (0.97–1.13)	2.54 (1.97–3.48) *	1.58 (1.22–1.66) ^†^	1.69 (1.20–1.78) ^†^
TC (mmol/L)	2.59 (2.59–2.60)	3.25 (3.14–3.37) *	2.63 (2.61–2.64) ^†^	2.69 (2.66–2.78) ^†^
HDL-C (mmol/L)	1.98 (1.87–2.05)	0.87 (0.68–1.24) *	1.28 (1.24–1.34)	1.30 (1.29–1.39)
Initial Body Weight (g)	240.7 ± 18.62	258.9 ± 37.52	271.5 ± 28.89	265.7 ± 43.99
Final Body Weight (g)	385.9 ± 33.75	509.0 ± 48.85 *	437.4 ± 21.80 ^†^	457.7 ± 59.02
PWG (%)	60.3 ± 8.60	98.0 ± 13.12 *	62.0 ± 9.63 ^†^	73.4 ± 13.91 ^†^
BMI (g/cm^2^)	0.71 ± 0.04	0.87 ± 0.03 *	0.74 ± 0.02 ^†^	0.78 ± 0.03 ^†^
AI (%)	3.76 ± 0.75	8.91 ± 0.56 *	4.87 ± 0.46 ^†^	5.51 ± 1.17 ^†^

Serum lipid profile data are expressed as median (percentile 25–75) (non-parametric k-independent median test). Anthropometric data are expressed as mean ± standard deviation (one-way ANOVA; post hoc Bonferroni). Values showing * *p* < 0.05 vs. C; ^†^ *p* < 0.05 vs. D. Abbreviations: C, control rats fed standard chow pellet diet and water; D, HFS diet-induced T2D rats; DE, HFS diet-induced T2D rats exposed to EE; DM, HFS diet-induced T2D rats treated with metformin; HDL-C, high-density lipoprotein cholesterol; TC, total cholesterol; TG, triglyceride; PWG, percentage of weight gain; BMI, body mass index; AI, adiposity index.

**Table 4 biology-12-00480-t004:** Effects of EE exposure and metformin treatment on oxidative status in HFS diet-induced T2D rats.

Parameters/Group	C	D	DE	DM
FORT (mmol H_2_O_2_ Eq/L)	2.03 ± 0.10	2.91 ± 0.18 *	2.49 ± 0.03 ^†^	2.49 ± 0.09 ^†^
FORD (mmol/L Trolox Eq/L)	1.43 ± 0.16	1.43 ± 0.15	1.64 ± 0.24	1.44 ± 0.17
Redox balance	0.71 ± 0.05	0.49 ± 0.03 *	0.66 ± 0.09 ^†^	0.58 ± 0.05

Data are expressed as mean ± standard deviation (one-way ANOVA; post hoc Bonferroni). Values showing * *p* < 0.05 vs. C; ^†^
*p* < 0.05 vs. D. Abbreviations: C, control rats fed standard chow pellet diet and water; D, HFS diet-induced T2D rats; DE, HFS diet-induced T2D rats exposed to EE; DM, HFS diet-induced T2D rats treated with metformin; FORD, free oxygen radicals defense test; FORT, free oxygen radicals testing.

**Table 5 biology-12-00480-t005:** Effects of EE exposure and metformin treatment on T-maze tasks and serum BDNF in HFS diet-induced T2D rats.

Parameters/Group	C	D	DE	DM
Spontaneous alternation (%)	84.52 ± 6.234	67.86 ± 4.64 *	85.12 ± 5.30 ^†^	85.71 ± 5.82 ^†^
Correct response (%)	86.31 ± 5.75	67.26 ± 3.75 *	80.36 ± 4.64 ^†^	79.76 ± 2.87 ^†^
Serum BDNF (ng/mL)	14.45 ± 3.43	9.55 ± 1.67 *	14.89 ± 1.00 ^†^	15.35 ± 2.27 ^†^

Data are expressed as mean ± standard deviation (one-way ANOVA; post hoc Bonferroni). Values showing * *p* < 0.05 vs. C; ^†^ *p* < 0.05 vs. D. Abbreviations: C, control rats fed standard chow pellet diet and water; D, HFS diet-induced T2D rats; DE, HFS diet-induced T2D rats exposed to EE; DM, HFS diet-induced T2D rats treated with metformin; BDNF, brain-derived neurotrophic factor.

## Data Availability

Not applicable.

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
