# Peer review of "Environmental Enrichment and Metformin Improve Metabolic Functions, Hippocampal Neuron Survival, and Hippocampal-Dependent Memory in High-Fat/High-Sucrose Diet-Induced Type 2 Diabetic Rats"

_biology, 2023, doi:10.3390/biology12030480_

Round 1
Reviewer 1 Report
In this study, the authors compare the effects of metformin, a drug of choice to treat type 2 diabetes mellitus (T2D), to the ones of environmental enrichment (EE) on metabolic dysfunctions, oxidative stress, hippocampal neuron death, and hippocampal-related memory defects in a rat model of T2D induced by an high-fat/high sucrose diet, mimicking Western-style diet. Very interestingly, EE showed similar anti-diabetic properties compared to the drug in this animal model.
In my opinion, the work has been clearly exposed, experimental design is good and the results are compelling. I have only minor considerations to express.
- Results would be clearer if expressed as box-and-whisker plots (or histograms that show individual values) instead of tables only.
- How randomized allocation into experimental groups has been performed? Please specify it in the material and methods section.
- Please specify the experiments for which blind analysis has been performed.
- Have the data been subjected to normality and equal variance analysis? How you decided to use Kruskal-Wallis or ANOVA tests?
Author Response
Dear Reviewer 1,
Thank you for your constructive feedback and comments We have addressed all your concerns in the attached document

Reviewer 2 Report
In the manuscript titled “Environmental enrichment and metformin improve metabolic functions, hippocampal neuron survival, and hippocampal-dependent memory in high-fat/high sucrose diet-induced type 2 diabetic rats” by Narendra Pamidi et al., The aim of this study is to explore the impact of environmental enrichment (EE) and metformin treatments on metabolic dysfunctions, neuronal death in the hippocampus, and memory impairments related to the hippocampus in rats with type 2 diabetes (T2D) induced by a high-fat/high-sucrose (HFS) diet.
The study has been well-designed, and the publication presents the results and experimental procedures in a clear manner. While the overall findings are not new, the study's identification of upregulated genes is a unique contribution. However, to strengthen this finding, Western blotting evidence would be necessary.
Specific comments:
1) Western blotting supporting altered expression of genes from the mouse tissues.
Author Response
Dear Reviewer 2,
Thank you for your constructive comments. We have responded to your concerns in the document attached.
